# Custom-Designed Portable Potentiostat and Indirect Cyclic Voltammetry Index Analysis for Rapid Pesticide Detection Using Molecularly Imprinted Polymer Sensors

**DOI:** 10.3390/s25102999

**Published:** 2025-05-09

**Authors:** Min-Wei Hung, Chen-Ju Lee, Yu-Hsuan Lin, Liang-Chieh Chao, Kuo-Cheng Huang, Hsin-Yi Tsai, Chanchana Thanachayanont

**Affiliations:** 1National Center for Instrumentation Research, National Institutes of Applied Research, Hsinchu 300092, Taiwan; niiat@niar.org.tw (M.-W.H.); chengru@niar.org.tw (C.-J.L.); andy556644@niar.org.tw (L.-C.C.); huangkc@niar.org.tw (K.-C.H.); kellytsai@niar.org.tw (H.-Y.T.); 2National Metal and Materials Technology Center, Pathum Thani 12120, Thailand; chanchm@mtec.or.th

**Keywords:** electrochemical sensor, cyclic voltammetry, molecularly imprinted polymer, pesticide detection, miniaturized device, characteristic index

## Abstract

Water pesticide contamination represents a major threat to ecological systems and public health, particularly in agricultural regions. Although conventional detection methods such as liquid chromatography and electrochemical analysis are highly accurate, they are expensive, require skilled operators, and cannot provide real-time results. This study developed a portable miniaturized electrochemical analysis platform based on cyclic voltammetry (CV) for rapid pesticide detection. The platform was compared with a commercial electrochemical analyzer and yielded similar performance in detecting chlorpyrifos at different concentrations. When ultrapure water was used as the background solution, the total area under the CV curve exhibited a linear correlation (*R*^2^ = 0.89) with the pesticide concentration, indicating its potential as a characteristic index. When molecularly imprinted polymers were added, the platform achieved a limit of detection of 50 ppm, with the area under the CV curve maintaining a logarithmic linear relationship (*R*^2^ = 0.98) with the pesticide concentration. These findings confirm the total area under the CV curve as the most reliable characteristic index for pesticide quantification. Overall, the proposed platform offers portability, straightforward operation, cost-effectiveness, and expandability, making it promising for on-site environmental monitoring. By incorporating GPS functionality, the platform can provide real-time pesticide concentration mapping, supporting its use in precision agriculture and water quality management.

## 1. Introduction

Heavy industries typically contribute to water pollution. At least two billion individuals consume water containing pollutants, including pesticides, heavy metals, and organic substances [1,2]. Monitoring water turbidity, pH, temperature, and other parameters is essential for maintaining water quality and protecting the environment. Various monitoring technologies have recently incorporated into Internet of Things modules [3,4]. Traditional water quality monitoring methods involve collecting water samples and analyzing them in a laboratory with advanced instruments to measure parameters such as pH, conductance, turbidity, and dissolved oxygen content. Kruse [5] comprehensively reviewed several water quality sensors and identified key chemical parameters that determine the potability of water. The complexity and high costs of sensors limit their widespread adoption. Electrochemical analysis is a commonly used sensing technique for determining concentrations of specific substances. Cyclic voltammetry (CV) is an important part of electrochemical analysis. Apetrei et al. [6] developed a multisensory system to analyze the natural waters of the Danube near the city of Galați. The system comprises an array of screen-printed sensors made from electroactive compounds and nanomaterials and exemplifies the use of advanced sensing technology in environmental monitoring. They compared their voltametric readings with physicochemical analysis results using partial least squares regression and obtained correlation coefficients exceeding 0.9. Gruden et al. [7] developed a CV-based sensor to determine the optimal dosage of detergent and demonstrated that factors such as water hardness, carbonate or noncarbonate hardness, and Ca/Mg ratio influenced the amount of detergent required. They also revealed that CV measurements can effectively differentiate between Ca/Mg ratio and carbonate hardness, thereby providing a reliable reference for detergent dosing, which can reduce environmental pollution. Carbó et al. [8] developed a voltametric electronic tongue to quantitatively evaluate spring water quality. This innovative system incorporated four noble metal electrodes housed within a stainless-steel cylinder and used pulse voltammetry to analyze 83 spring water samples. The system successfully determined nitrate, sulfate, fluoride, chloride, and sodium concentrations. Liao et al. [9] developed an integrated system consisting of a whole-copper electrochemical sensor, a handheld detector, a smartphone app, and a cloud-based mapping website for water quality assessment. The system was used to detect Pb^2+^ ions and determine chemical oxygen demand. Ramadhan et al. [10] used CV to determine the concentration of cobalt (Co) in contaminated water samples. To optimize the measurement conditions, they varied parameters such as voltage range, scan rate, and pH value. Subsequently, they validated their results against those of atomic absorption spectrometry and obtained a high correlation coefficient (0.9992). Aman et al. [11] used CV techniques involving a Pt-Cu electrode to selectively measure nitrate concentrations and estimate the corrosion rates of ions to determine the levels of contamination in the Buriganga River in Bangladesh, noting an accumulation of disintegrated organic pollutants on the riverbed.

Barvosa et al. [12] explored different voltametric methods used to detect pesticides and herbicides and highlighted the use of carbon electrodes and nanomaterials to achieve enhanced sensitivity and selectivity. They also reported the successful detection of contaminants in food samples at nanomolar levels, confirming the potential of voltametric sensors in environmental and food safety applications. Farouil et al. [13] used CV with immobilized particles to screen various pesticides and examined the effects of modifying certain characteristic signals on insecticides. They revealed that CV can effectively differentiate pesticides by their electrochemical behaviors, thereby providing valuable insights for the development of more efficient mosquito control strategies. Jevtić et al. [14] reviewed recent advancements in electrochemical techniques and reported that electrochemical methods offer rapid analysis, simplicity, cost-effectiveness, and minimal sample preparation. However, they also discussed the limitations associated with the lack of electrochemical activity of many pesticides and highlighted the use of various electrode materials and modifiers to enhance the performance and applicability of electrochemical sensors for pesticide analysis. Noori et al. [15] provided an overview of advancements in the electrochemical detection of pesticides such as glyphosate, lindane, and bentazone. They highlighted the use of various electrochemical detection techniques, electrode materials, and electrolyte media to enhance the sensitivity and selectivity of pesticide detection and confirmed the ability of novel electrode materials to achieve low detection limits and broad linearity ranges, showcasing their potential in real-world applications. Table 1 presents a comparison of various electrochemical techniques, including square wave voltammetry, differential pulse voltammetry, and chronoamperometry. This comparison involves parameters such as sensitivity, detection limit, time efficiency, and applicability to different pesticide concentrations. As shown in Table 1, CV represents a comprehensive and flexible technique for measuring pesticide concentrations. Its moderate sensitivity, broad applicability, and ease of interpretation make it an optimal approach for routine analysis in both laboratory and field settings.

Silva et al. [16] researched the development and application of an electrochemical sensor for pesticide detection and used screen-printed electrodes modified with various nanomaterials to enhance the sensor’s sensitivity and selectivity. They also used CV to detect electrochemical signals corresponding to different pesticides. By integrating principal component analysis and artificial neural network classification, they accurately identified and quantified different pesticides in an environmental sample. Molecularly imprinted polymers (MIPs) serve as highly selective and sensitive recognition elements in electrochemical sensors. These synthetic polymers are engineered to possess specific binding sites that match the shape, size, and functional groups of certain target analytes, which in turn provide them with excellent specificity. This high specificity enhances the detection capabilities of electrochemical sensors, making them optimal tools for identifying trace amounts of contaminants, such as pesticides or pharmaceuticals, in complex matrices. MIPs exhibit high stability and durability and can be repeatedly used without any major loss in performance. These tools’ robustness and cost-effectiveness make them a practical alternative to biological recognition elements, thereby offering a reliable and economical solution for environmental monitoring, food safety, and medical diagnosis. Integrating MIPs into electrochemical sensors improves overall sensitivity and accuracy, ensuring that sensors can precisely quantify analytes in diverse applications. Aghoutane et al. [17] developed an MIP-based electrochemical sensor for the detection and determination of malathion, an organophosphorus insecticide that poses substantial health and environmental risks, in olive oils and fruits. They reported that their sensor achieved high selectivity and sensitivity and had a low limit of detection (LOD); their sensor had a LOD of 0.06 pg/mL. Mugo et al. [18] developed a portable electrochemical sensor to detect imidacloprid, a widely used neonicotinoid pesticide, that was equipped with microneedle technology and MIP modifications to enhance its selectivity and sensitivity. The sensor detected imidacloprid at concentration ranges of 2.0–99 µM using CV and 0.20–92 µM using differential pulse voltammetry, with LODs of 0.35 and 0.06 µM, respectively. The sensor also demonstrated excellent reusability and successfully quantified imidacloprid in honey samples.

Traditionally, metal or plastic electrodes interact with other substances only on their surface, which represents a challenge in cases of low-concentration pollutants. Additionally, the multifunctionality and high cost of commercial electrochemical instruments limit their use, particularly in developing countries. Table 2 provides a summary of the characteristics of these instruments. Li et al. [19] developed a paper-based electrode coated with nanomaterials to detect pollutants in water. They also developed a portable CV measurement tool to rapidly obtain information on oxidation and reduction currents and corresponding potential voltages from different concentrations of electrolytes and specific chemical substances. In their study, they compared the results of commercial plastic and self-printed paper-based electrodes with those of commercial and self-developed electrochemical modules. They reported that the performance of the self-developed electrochemical modules was identical to that of the standard commercial instruments. Electrodes coated with shell-less MIPs typically possess a 1.5- to 2-fold higher current variation for chlorpyrifos detection compared with uncoated electrodes. CV indicators play a crucial role as a reference for evaluating test sample concentrations. Chen et al. [20] developed a novel data analysis and CV technique to determine the concentration of carbofuran, a pesticide. They identified peak currents and specific area indices from CV curves as key dependent variables, with pesticide concentration serving as the independent variable. They reported that the CV area between 0.5 and 0.9 V served as the most precise index for pesticide concentration, achieving an accuracy of 5 ppm with an inverse calibration equation.

Due to the considerable complexity of its electrochemical response when interacting with the modified electrode and the matrix environment, the electrochemical signals are influenced not only by the intrinsic properties of the pesticide but also by the MIP layer and potential secondary processes occurring within the electrolyte system. Therefore, this study developed an analytical platform that comprises a self-developed electrochemical analyzer and four analytical indices. These indices are the area under the CV curve, the reduction and oxidation peak current, the potential at peak current, and the instantaneous current at the highest scan potential. We analyzed four concentrations of chlorpyrifos in different background solutions (an electrolyte solution and ultrapure water). We then established a relationship between pesticide concentration and these four indices to identify the key index for accurate detection.

## 2. Materials and Methods

### 2.1. Cyclic Voltammetry

Cyclic voltammetry (CV) is an electrochemical technique in which the potential of the working electrode is linearly varied with time in a cyclic manner. Unlike linear sweep voltammetry, in which the potential increases only in a single direction, CV involves sweeping the potential to a peak voltage and then reversing it to its initial value, achieving a full cycle. This cycle is typically repeated two or three times to achieve data stability, allowing reactions at the electrolyte and electrode surface to reach equilibrium. To obtain a cyclic voltammogram, the current measured from the working electrode and counter electrode is plotted against the applied potential voltage. This plot demonstrates the electrochemical properties of the test solution by illustrating the oxidation and reduction currents and their corresponding potentials. Elgrishi et al. [21] and Joshi et al. [22] comprehensively explained the principles of CV and how to interpret and classify voltammograms. Voltammograms are interpreted in accordance with United States conventions or International Union of Pure and Applied Chemistry conventions. These conventions differ in potential scanning direction. Typically, the peaks in a cyclic voltammogram indicate the oxidation and reduction events of the analytes in the solution. These events can be used to determine electrochemical behaviors (Figure 1).

CV measurements enable the analysis of various characteristic indices and serve as a reference for examining samples of different concentrations. Indices include the total area under the CV curve, the peak currents of oxidation and reduction, the potential at peak current, and the instantaneous current at the highest scan potential. The total area under the CV curve represents the overall charge that passes during an electrochemical reaction. It reflects the total electroactive species present in the solution and is used to quantify the extent of the redox processes that occur on the electrode’s surface. Oxidation peak current (*I_pa_*) and reduction peak current (*I_pc_)* provide insights into the rate of electron transfer reactions and the concentration of analytes in solution. The voltages at which these peak currents occur (*E_pa_* for oxidation and *E_pc_* for reduction) indicate the potential at which the oxidation or reduction of the analyte begins, reflecting the energy required for these processes. The instantaneous current at the highest scan potential (*I_hv_*) serves as a sensitive electrochemical parameter for quantifying the concentration of redox-active species in a solution. This parameter reflects the efficiency of the electron transfer process and the electrochemical activity of the test species. Collectively, these indices provide valuable insights into the electrochemical properties and behaviors of the analyte and aid in the characterization and quantification of the substances involved in electrochemical reactions. This study explored the relationships between various characteristic indices and sample concentrations and identified highly correlated characteristic factors as a reference for future applications.

### 2.2. Molecularly Imprinted Polymers (MIPs)

Molecularly imprinted polymers (MIPs) are synthetic receptors engineered to recognize specific target analytes. MIPs contain cavities with a high affinity for the template analytes used during their synthesis (Figure 2). During polymerization, the size, shape, and charge distribution of these cavities are modified to ensure high specificity and affinity for the target analytes, such as specific substances found in pesticides. Consequently, MIPs are extensively used as chemical separators and molecular sensors. Leepheng et al. [23] developed a smartphone-based pesticide residue detector that integrates an MIP-modified electrode with a near-field communication sensor. In this system, the pesticide binds to MIPs fabricated onto screen-printed electrodes to create detectable cavities on the surface. This integrated detector, combined with MIP-modified electrodes, can specifically detect pesticides such as cypermethrin and carbaryl in vegetables and fruits.

When applied to electrodes in CV measurements, these cavities provide a highly selective recognition site for target analytes, ensuring that the electrochemical signals observed are predominantly attributable to the presence of these specific analytes. MIPs usually result in a pronounced peak current and a clear signal at the corresponding potential voltages. They also contribute to an increase in the signal-to-noise ratio, which, in turn, facilitates the detection and quantification of target analytes at low concentrations.

### 2.3. Experimental Setup and Samples

#### 2.3.1. Potentiostat and Sensor

This study designed a CV control and measurement circuit. This measurement circuit incorporated a digital-to-analog conversion and CV, resulting in a simple, compact, and miniaturized module measuring 8 × 8 × 3 cm^3^ (Figure 3). Unlike high-end, precision commercial analytical instruments, which are large and involve complex operating procedures, this module was portable and easy to use, which made it suitable for rapid experimentation in different locations.

The screen-printed carbon electrode (SPCE) comprised a working electrode and a counter electrode fabricated using carbon-based conductive ink, along with a reference electrode printed with silver/silver chloride (Ag/AgCl) ink. The electrode layout was defined and insulated by a polymeric layer to ensure precise control over the active surface areas. The working electrode featured a circular geometry with a diameter of 3.35 mm.

#### 2.3.2. Electrochemical Measurements

In the experimental setup, a commercial electrochemical analyzer from CH Instruments (Model 600E series, CHI, Bee Cave, TX, USA) was used for benchmarking. The software was CHI 6124E (CHI Version 24.01). This simplified CV control and measurement circuit, along with a screen-printed three-electrode sensor system, is depicted in Figure 4. The standard electrolyte used for system comparison consisted of potassium hexacyanoferrate(II) trihydrate (K_4_[Fe(CN)_6_]·3H_2_O) and potassium ferricyanide crystals (K_3_Fe(CN)_6_) at concentrations of 0.01, 0.02, and 0.03 mol/L (M). Converting these concentrations, they are equal to 3476, 6928, and 10,356 ppm, respectively. Each electrode was covered with 50 μL of this solution. The potential voltage (*E*) applied between the working electrode and reference electrode ranged from −1 to +1 V, with a voltage increment of 5 mV. In the circuit design and experimental process, the primary consideration was to compare the time required for each cycle within the voltage range of −1 to +1 V, aiming to achieve a measurement duration comparable to that of the CHI system. When the step size is excessively small, the time consumed per cycle increases significantly. Conversely, employing a larger step size substantially shortens the measurement time but compromises the resolution of the acquired data. Therefore, a step size of 5 mV was selected as an optimal compromise, balancing measurement duration and resolution for reliable and efficient analysis. The current flowing to or from the working electrode was calculated from the voltage measured across a resistor. These values were then used to calculate the current at different potential voltages. In response to the different concentrations of electrolytes or pesticides, the resistance values were adjusted to ensure that the measurement range remained within the operational limits of the instrument. The resistance values used in the experiment ranged from 25 to 1000 Ω.

#### 2.3.3. MIP Synthesis and Modification

Functional monomers (methacrylic acid), cross-linker (ethylene glycol dimethacrylate), and initiator (azobisisobu-tyronitrile) were prepared to create the MIPs with the chlorpyrifos acting as the target molecule. The process involved stirring, heating, and maintaining at a specific temperature for 24 h, followed by washing and drying, resulting in what we refer to as “shelled” MIPs. After removing the template molecule, the MIPs became “shell-less”. The prepared MIP samples are dissolved in 1 mL of chitosan solution (0.2% wt, pH 6.0) and then coated onto the electrode for testing.

The molecularly imprinted polymer (MIP) specific to chlorpyrifos was synthesized using a bulk polymerization approach. Initially, chlorpyrifos, serving as the template molecule, was dissolved in acetonitrile and transferred into a three-neck round-bottom flask. Subsequently, methacrylic acid (MAA), the functional monomer, was added to the mixture. The solution was stirred for an additional 60 min to allow non-covalent interactions between the monomer and the chlorpyrifos molecules. Following this, ethylene glycol dimethacrylate (EGDMA), a cross-linking agent, was introduced into the mixture, which was then stirred for another 60 min at room temperature. Azobisisobutyronitrile (AIBN), a free-radical initiator, was subsequently added. The mixture was purged with nitrogen gas for 10 min to remove dissolved oxygen and prevent inhibition of polymerization. The polymerization reaction was then carried out at 65 °C for 24 h. After polymerization, the solid polymer was collected and washed with ethanol to remove any unreacted monomers and impurities. The embedded chlorpyrifos molecules were removed using Soxhlet extraction with a methanol:acetic acid mixture (90:10 *v*/*v*) for 6 h, leaving behind specific binding sites complementary to the shape and functional groups of chlorpyrifos. To ensure neutrality, the polymer was washed with deionized water until a neutral pH was reached. Finally, the polymer was dried in an oven at 60 °C for 12 h and stored in a desiccator to maintain its stability and prevent moisture absorption when not in use.

#### 2.3.4. Pesticide

Chlorpyrifos was the target pesticide. This pesticide was prepared at a concentration of 20% (200,000 ppm) for electrochemical analysis. Chlorpyrifos is an organophosphorus insecticide with the chemical formula C_9_H_11_Cl_3_NO_3_PS. Its chemical structure involves two key components: a pyridyl group (3,5,6-trichloropyridinyl) and a thionophosphate group (O,O-diethyl thiophosphate). In our experiment, 50 μL of ultrapure water or an electrolyte solution consisting of potassium hexacyanoferrate(II) trihydrate and potassium ferricyanide crystals were used as the background solutions, and 50 μL of a pesticide sample was used as the target analyte for CV measurement. Pesticide samples were diluted to 50,000, 100,000, and 150,000 ppm to determine the variations in their CV curve characteristics and establish a correlation between CV curve characteristics and pesticide concentration. Because the test sample was synthetic and did not contain other substances, we assumed that our experiment was not influenced by any other compounds. Additional interference studies focusing on samples collected from river environments are required for comparison.

### 2.4. Experimental Procedure

Overall, our experimental procedure involved six steps aimed at measuring the CV curves of pesticide solutions and analyzing characteristic indices such as the total area under the CV curve, peak current of oxidation or reduction, voltage at peak current, and instantaneous current at the highest scan potential at different pesticide concentrations. During the experiment, each pesticide sample was measured twice, with each measurement consisting of three CV scans, resulting in a total of six data sets per concentration. The first cycle of each measurement was excluded because of the potentially incomplete wetting between the sensing solution and the MIP-coated electrode, with a deviation in curve trends from subsequent cycles. The remaining four cycles exhibited uniform responses. Therefore, a single representative data point was selected for concentration analysis. The coefficient of determination (*R*^2^) was then used to evaluate the reliability of different characteristic indices as indicators of pesticide concentration. This process was intended to establish the most suitable characteristic indices for the rapid measurement and analysis of pesticide concentration. A flowchart of the experimental procedure is given in Figure 5. Each step is described in detail in the following paragraphs.

Step I: After a CV control circuit was developed, a miniaturized electronic module was constructed. This CV module served as a potential sweep module and incorporated three electrodes: a counter electrode, a working electrode, and a reference electrode. It also linearly swept the potential (*E*) between the reference electrode and working electrode over time and simultaneously measured the current flow between the counter electrode and working electrode. The desired potential was achieved by automatically regulating the voltage across the circuit, from the counter electrode to the working electrode, through closed-loop control.

Step II: Electrodes on screen-printed plastic substrates, along with a standard electrolyte, were prepared. Before the analysis of pesticide samples, the reduction and oxidation currents and the corresponding potential voltages of the electrolyte were recorded. Three smaller electrodes were then prepared on plastic-based substrates. For these substrates, carbon ink was used as the working electrode and counter electrode, and silver or silver chloride ink was used as the reference electrode.

Step III: To verify the accuracy of the self-developed CV module, standard electrolyte tests were conducted using screen-printed plastic substrates, followed by CV curve measurements with both commercial instruments and the self-developed module. In addition, electrolyte solutions of varying concentrations were examined to determine the relationship between electrolyte concentration and the corresponding reduction and oxidation current values. Finally, an optimal electrolyte concentration was selected to serve as the background solution for subsequent pesticide sample testing.

Step IV: MIPs were prepared in powder form and dissolved in water to obtain a solution. This solution was then applied to the working electrode of a screen-printed electrode on the substrate. After approximately 10 min of air-drying, the MIPs were affixed to the working electrode to capture components from the pesticide sample and enhance the measurement signal.

Step V: Pesticide samples containing chlorpyrifos were attached to the screen-printed electrode on a plastic-based substrate. The background solution was an electrolyte solution or ultrapure water. A cyclic voltammogram of the pesticide samples was then obtained using the self-developed CV module. Pesticide samples of varying concentrations were used to measure their CV curves.

Step VI: The total area under the CV curve, peak current, potential at peak current, and instantaneous current at the highest scan potential associated with the pesticide samples were analyzed. Additionally, the optimal and most representative characteristic indices were identified for the rapid detection and analysis of pesticide concentration in the future.

## 3. Results and Discussion

The CV module developed in this study was compared with a commercial electrochemical analyzer (600E Series; CH Instruments). A standard electrolyte background solution and a standard pesticide sample were used in our experiments. Because the sensor may be used in field applications involving river water in which electrolytes may not be present, we also conducted experiments involving other background solutions. We also determined the module’s lowest LODs.

### 3.1. Comparison of Module with Commercial Instruments

This study compared the performance of the self-developed module (NCIR) with that of a commercial instrument (CHI). In this comparison, we used an electrolyte solution consisting of potassium hexacyanoferrate(II) trihydrate (K_4_[Fe(CN)_6_]·3H_2_O) and potassium ferricyanide crystals (K_3_Fe(CN)_6_). We then analyzed the CV curve and characteristic indices at three different concentrations (3476, 6928, and 10,356 ppm). Our findings revealed a uniform initial current measured by both instruments at low electrolyte concentrations (3476 and 6928 ppm), suggesting comparable performance (Figure 6). At an electrolyte concentration of 10,356 ppm, discrepancies in the initial current values were observed, which can be attributed to differences in equipment sensitivity, stability, and solution conductivity. The CV curves also revealed that the total area, peak reduction current (*I_pc_*), and potential at peak reduction current (*E_pc_*) increased with increasing electrolyte concentration. Overall, this trend suggests that high electrolyte concentrations enhance electrochemical activity, as indicated by an increase in peak current and an expansion in CV curve area. Peak-to-peak separation is a key parameter that provides valuable insights into the kinetics of electrochemical processes. For instance, the degree of peak-to-peak separation between oxidation and reduction peaks (Δ*E* = *E_pa_* − *E_pc_*) is often used to evaluate the reversibility of an electrochemical reaction. In such a scenario, when the concentration of the electrolyte increases, the degree of peak-to-peak separation is expected to change, suggesting a change in the electrochemical reaction’s dynamics. At low electrolyte concentrations (i.e., 3476 and 6928 ppm), the degree of peak-to-peak separation is relatively low, suggesting that the electrochemical process is reversible, with the system operating in a stable manner. By contrast, at high electrolyte concentrations (i.e., 10,356 ppm), the degree of peak-to-peak separation increases, likely because of an increase in solution conductivity, which, in turn, affects the diffusion rate of the analyte ions and influences the reaction kinetics. This discrepancy emphasizes the importance of carefully selecting an electrolyte concentration that optimizes the sensitivity and accuracy of measurements, particularly in the context of pesticide concentration analysis.

In this study, given the uniform performance observed at low concentrations and the need to minimize the influence of high electrolyte concentrations on the oxidation–reduction curve during pesticide concentration analysis, an electrolyte concentration of 3476 ppm was selected for the electrolyte background solution for subsequent experiments. This choice ensured reliable measurement results similar to those obtained with commercial instruments. It also reduced the potential interference from high electrolyte concentrations.

### 3.2. Variations in CV with Pesticide Concentration in Electrolyte

Upon selecting a 3476 ppm as the electrolyte concentration for the electrolyte background solution, the experiment proceeded with 50 μL of electrolyte background solution and 50 μL of pesticide, with a pesticide concentration ranging from 50,000 to 200,000 ppm. As shown in Figure 7, the peak currents for both oxidation and reduction decreased with the increasing pesticide concentration. However, their absolute values increased. Notably, the CV curve for the 150,000 ppm concentration deviated from the expected linear relationship, likely because of the nonuniform distribution of the MIPs and pesticide samples.

Pulse-like signals on CV curves are typically correlated with experimental factors related to scan rate, electrode surface, and electrochemical system stability. This correlation is primarily attributable to the discrete potential step resolution used during CV measurements. In this study, we increased the applied potential in increments of 5 mV, which may not be sufficient to generate a smooth CV curve, particularly when the electron transfer kinetics are not rapid enough to respond to the applied potential changes. This discrepancy may lead to a staircase-like or pulse-like appearance, particularly in cases involving nonoptimal electrochemical interfaces. Additionally, the observed pulse-like response may be influenced by the surface properties of the MIP-coated electrode. The presence of imprinted cavities on the electrode’s surface may lead to localized charge transfer phenomena, which may cause discontinuities in the recorded current signal, particularly in cases involving pesticides of varying concentrations. This effect may be amplified in the presence of nonuniform adsorption or partial blockage on the electrode’s surface by pesticide molecules, which alters the electron transfer process. In addition to the aforementioned effects, this entire process is influenced by the ionic strength and conductivity of the electrolyte solution. In this study, an electrolyte concentration of 3476 ppm was selected to ensure signal stability and minimize interference. At lower electrolyte concentrations, ionic conductivity may be suboptimal, leading to high solution resistance and potentially affecting current response, particularly during peak oxidation or reduction events. Additionally, the appearance of a minor anodic peak around 0.5 V may be attributed to the complex nature of the pesticide, whose molecular structure potentially interacts with the electrode surface, resulting in reversible or quasi-reversible redox processes. Additionally, variations in electrolyte composition, fluctuations in pH, and the presence of trace impurities could alter the electrochemical environment, thereby contributing to the emergence of secondary oxidation peaks within the cyclic voltammogram.

Further analysis of the CV curve (Figure 8) revealed a linear correlation between total area, *I_pc_*, and *I_pa_* with respect to the concentration of pesticide in the background solution. The linear correlation value (*R*^2^) for total area was calculated as only 0.66, whereas the linear correlation values for *I_pc_* and *I_pa_* were calculated as 0.72 and 0.76, respectively. Based on the experimental results, under the current conditions, the relatively low correlation between sample concentration and total voltametric area may be attributed to the response and sensitivity of the MIP towards the pesticide analyte. These values indicated moderate linearity, suggesting the reliability of CV curve area and peak current as indicators of pesticide concentration, albeit with certain variability. As shown in Figure 9, the linear correlation value for the oxidation potential was calculated as 0.85. This value indicates that, in the presence of an electrolyte background solution, the potential of the oxidation reaction is the most suitable characteristic index of pesticide concentration, with a linear correlation of more than 85%. Therefore, this value is regarded as the optimal choice for accurate and reliable pesticide concentration analysis. In contrast, the low *R*^2^ (0.03) of the linear fit indicates that *E_pc_* is essentially independent of concentration under the current conditions. Future work will focus on optimizing the design of the MIP and refining the electrode modification process, with the aim of enhancing the relationship between sample concentration and total CV area and potential corresponding to peak reduction current, thereby improving the sensor’s quantitative performance.

### 3.3. Variations in CV with Pesticide Concentration in Ultrapure Water

Using an electrolyte as a background solution in outdoor experiments is challenging. We next used ultrapure water as the background solution. Without an electrolyte present, distinguishing the peak currents of oxidation and reduction from CV curves is challenging. Therefore, we examined the correlation between pesticide concentration and the instantaneous current at the highest scan potential (*I_hv_*). We also measured the total area under the CV curve. If curve saturation occurred during measurement, the resistance in the self-developed module decreased. This effect typically expanded the measurement range but reduced the measurement’s resolution, resulting in a step-shaped CV curve. This setting accelerated the measurement process (Figure 10). In future studies, we intend to refine our measurement circuitry to enhance resolution and improve data accuracy.

As shown in Figure 11, the total area under the CV curve increased with increasing pesticide concentration, with a linear correlation value of *R*^2^ = 0.89. By contrast, the value of *I_hv_* demonstrated a weak linear correlation of approximately *R*^2^ = 0.69. Therefore, when analyzing the concentration of pesticide with ultrapure water as the background solution, the total area under the CV curve serves as a reliable characteristic index.

### 3.4. LODs of Pesticide Concentration with Self-Developed Module

Our experimental results indicate that the characteristic indices obtained from CV curves can be used to determine the concentrations of pesticide samples. In this study, to identify the LOD of the self-developed module with the current MIPs, the 5% (50,000 ppm) pesticide sample was diluted 10 times to determine the minimum detection limit. LOD refers to the lowest concentration or amount of analyte that can be reliably detected. Its value represents the minimum concentration at which the analyte can be distinguished from the background signal. LOD can be determined through various approaches, including statistical methods, signal-to-noise ratio analysis, and visual determination. In this study, we adopted the visual determination method, in which LOD is defined as the lowest concentration at which a measurable signal can be correlated with analyte concentration. This approach was used to establish the detectable range of visual determination.

As shown in Figure 12, at pesticide concentrations of 50–50,000 ppm, the total area under the CV curves exhibited a linear relationship with the logarithm of pesticide concentration, with a high *R*^2^ value (0.98). In addition, the LOD of the current configuration for chlorpyrifos was calculated to be 50 ppm. Identifying the correlation between current at peak voltage and pesticide concentration is challenging. Therefore, this module can be paired with MIP-equipped sensors to detect and screen various pesticides at different concentrations. We intend to conduct additional experiments in future studies to determine the limit of quantification, evaluate precision through repeated measurements across different days and conditions, and assess sensitivity by performing analyses involving smaller concentrations of target analytes. These modifications are expected to facilitate more accurate and reliable pesticide detection in diverse environmental samples.

## 4. Conclusions

This study developed an electrochemical analysis platform that incorporated a self-developed electrochemical analyzer and characteristic indices for the detection of pesticides at various concentrations. Initially, we examined the characteristic indices of the total area under the CV curve, the peak currents of oxidation and reduction, the potential at peak current, and the instantaneous current at the highest scan potential. We then used these indices to establish a correlation with pesticide sample concentration and to determine the LOD. We used MIP-coated electrodes to determine the concentration of chlorpyrifos. Our experimental results indicated that the performance and measurement trends of our self-developed electrochemical analyzer were highly similar to those of commercial instruments. Our results also revealed that, in the electrolyte background solution, the characteristic index of potential corresponding to the oxidation peak current exhibited a strong correlation of 85% with concentration. When the module was combined with MIPs, it achieved an LOD of 50 ppm for chlorpyrifos. Analysis of the total area of the CV curve revealed a correlation coefficient exceeding 95%. In the future, we intend to explore additional strategies for enhancing correlation coefficients, such as optimizing electrode surface modifications, refining experimental conditions, and improving background electrolyte compositions. Overall, optimizing MIP samples and modules may further reduce the detection limit to the part-per-billion level. Additionally, integrating a GPS module to obtain information on pesticide concentration distribution across river basins may facilitate environmental monitoring and precision agriculture.

## Figures and Tables

**Figure 1 sensors-25-02999-f001:**
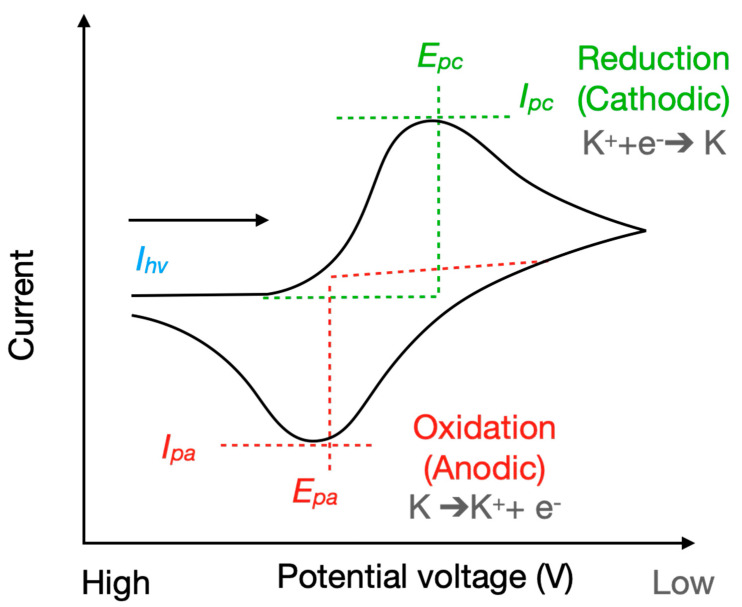
Curve obtained from CV measurements (per United States’ conventions) [21].

**Figure 2 sensors-25-02999-f002:**
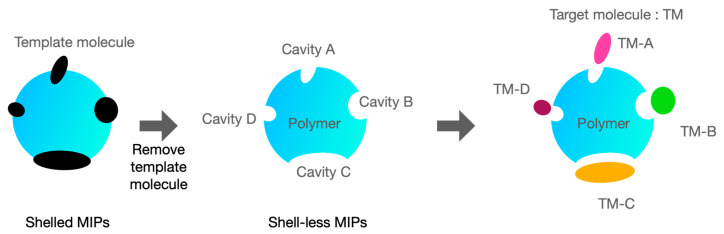
MIPs for capturing target analytes.

**Figure 3 sensors-25-02999-f003:**
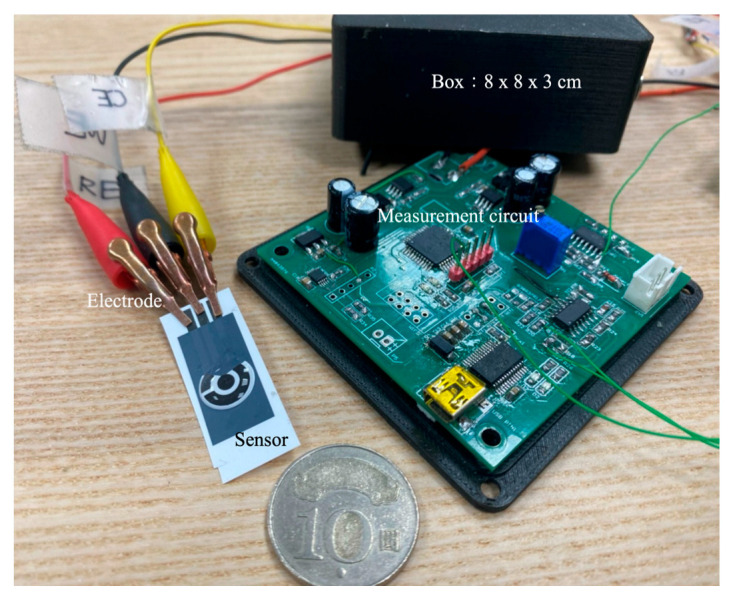
Self-developed CV module.

**Figure 4 sensors-25-02999-f004:**
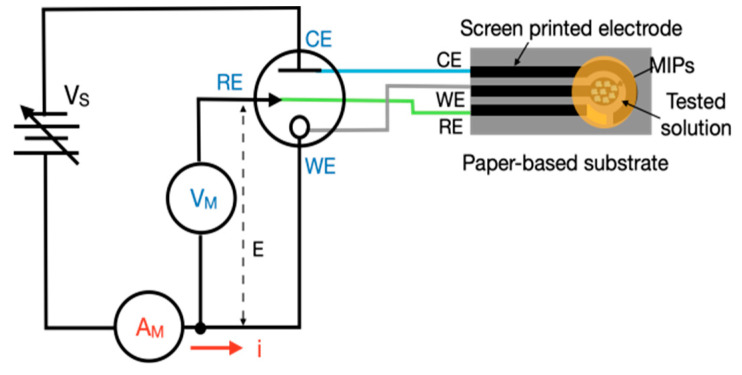
Schematic of simplified CV control and measurement circuit and three-electrode sensor with MIPs for solution testing.

**Figure 5 sensors-25-02999-f005:**
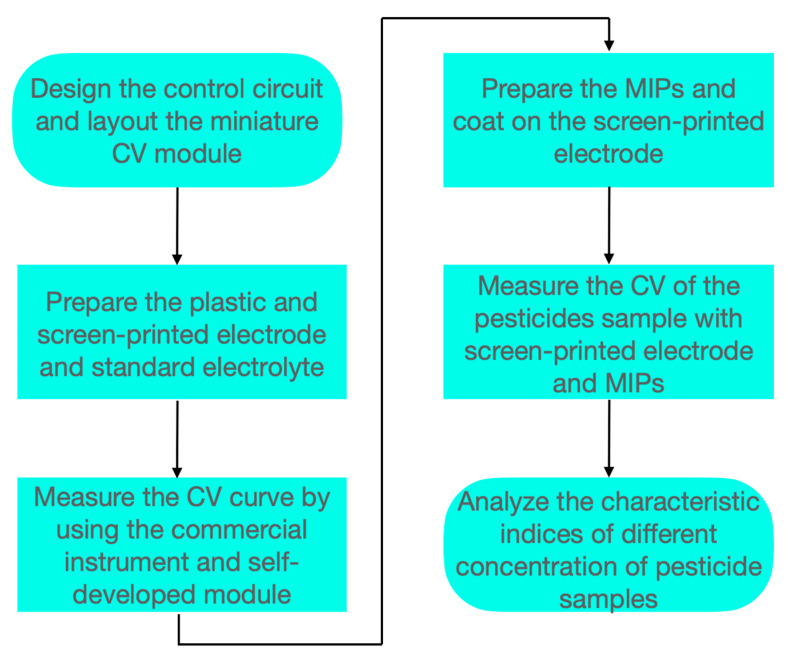
Experimental procedure.

**Figure 6 sensors-25-02999-f006:**
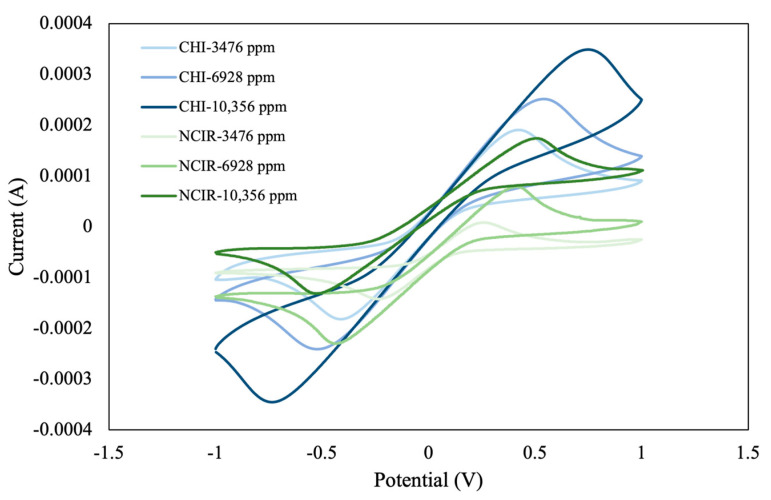
CV curves of potassium hexacyanoferrate(II) trihydrate and potassium ferricyanide crystals at concentrations of 3476, 6928, and 10,356 ppm, measured using the self-developed module and commercial instrument.

**Figure 7 sensors-25-02999-f007:**
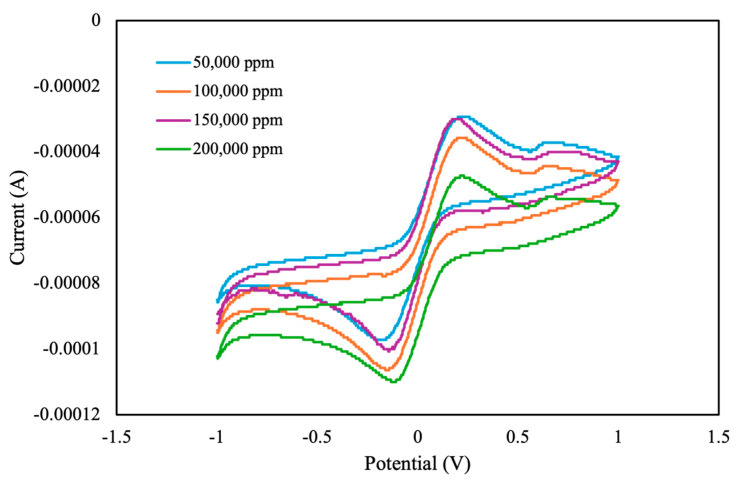
CV curves of different pesticide concentrations in electrolyte solution, measured using the self-developed module.

**Figure 8 sensors-25-02999-f008:**
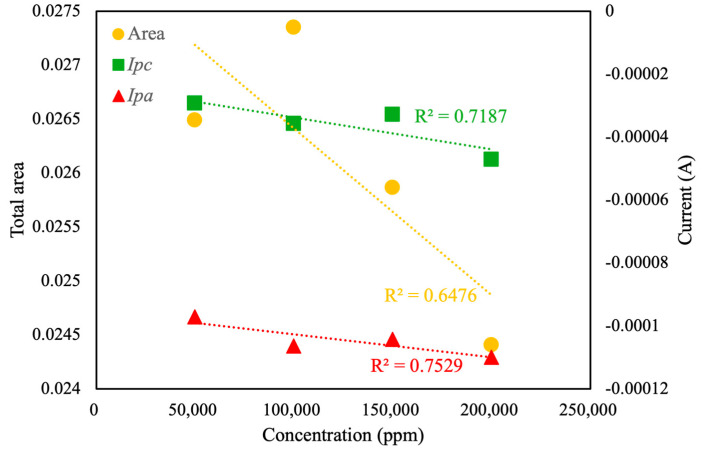
Total areas and *I_pc_* and *I_pa_* values of different pesticide concentrations in electrolyte solution, measured using the self-developed module.

**Figure 9 sensors-25-02999-f009:**
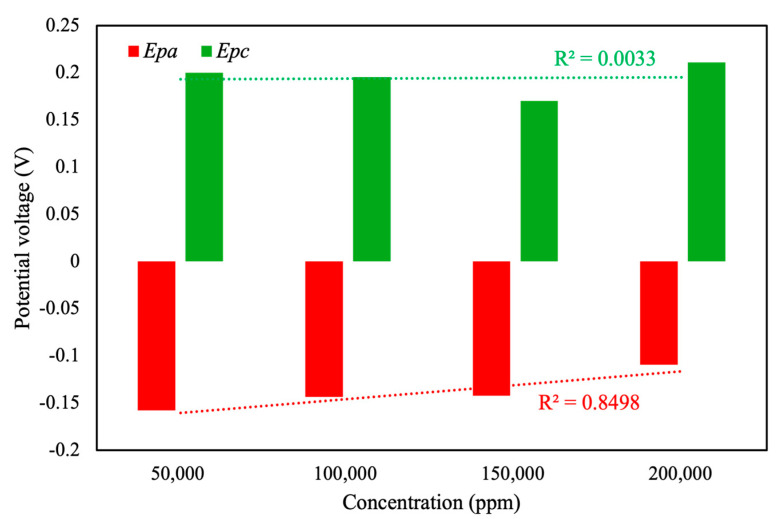
*E_pc_* and *E_pa_* values of different pesticide concentrations in electrolyte solution, measured using the self-developed module.

**Figure 10 sensors-25-02999-f010:**
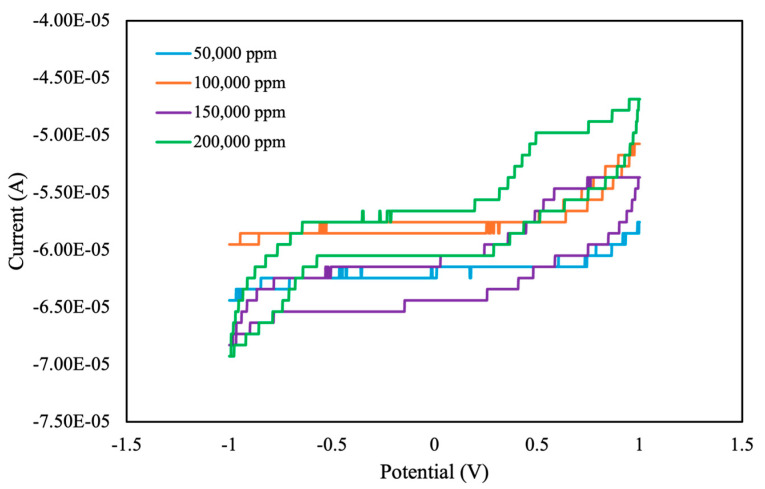
CV curves of different concentrations of pesticide in ultrapure water, measured using the self-developed module.

**Figure 11 sensors-25-02999-f011:**
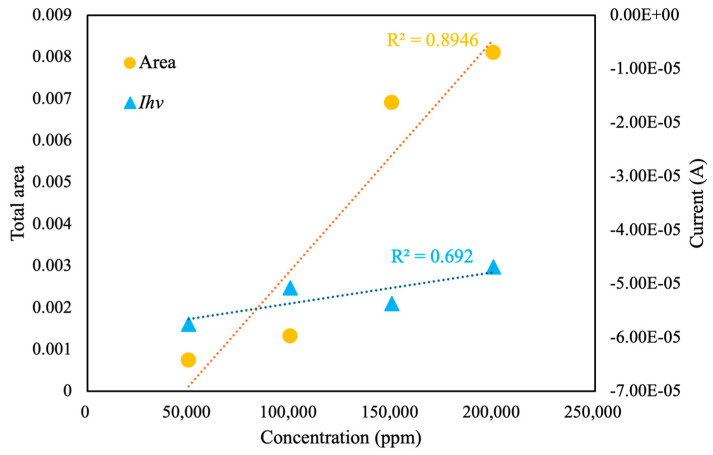
Total area and instantaneous current at highest scan potential (*I_hv_*) of different concentrations of pesticide in ultrapure water, measured using the self-developed module.

**Figure 12 sensors-25-02999-f012:**
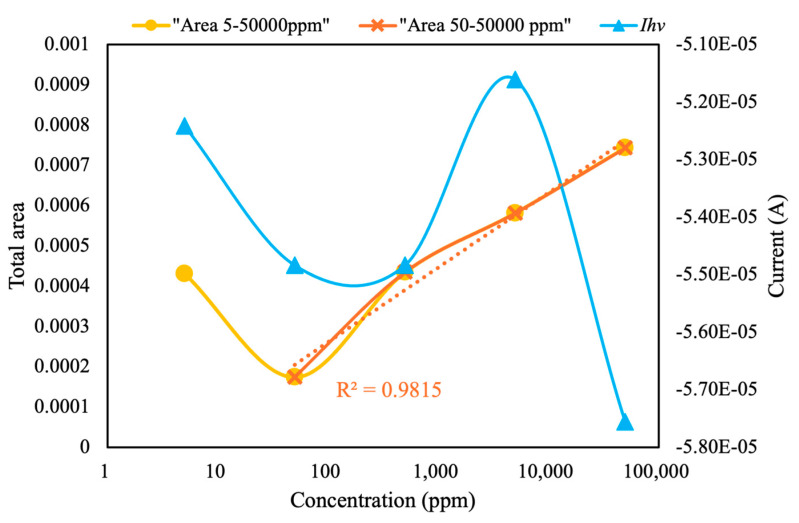
LODs of pesticide from total area and current at peak voltage (*I_hv_*) in ultrapure water, measured using the self-developed module.

**Table 1 sensors-25-02999-t001:** Comparison of different electrochemical techniques.

Technique	Sensitivity	Detection Limit	Time Efficiency	Advantages	Disadvantages
Cyclic Voltammetry (CV)	Moderate	Higher limits	Moderate	Suitable for broad analysis, provides full current-voltage response.	Time-consuming, may require complex data interpretation
Square Wave Voltammetry (SWV)	High	Low limits	Fast	High sensitivity, minimal background interference, faster than CV.	Limited to specific analytes and conditions.
Differential Pulse Voltammetry (DPV)	High	Low limits	Moderate	High sensitivity, suitable for complex samples.	May suffer from signal interference.
Chronoamperometry (CA)	Moderate	Moderate to low	Very fast	Simple, fast, provides information on diffusion processes.	Limited to diffusion-controlled reactions.

**Table 2 sensors-25-02999-t002:** Comparison of electrochemical analyzers.

Device	Characteristics
Commercial Instruments	High accuracy Multi-functions Large device dimension Use in the laboratory High cost of device (USD 10,000–20,000)
Self-Developed (NCIR) Module	High accuracy Single function (can be expended) Small device dimension Use in the laboratory and outdoor Low cost of device (USD 500–1000)

## Data Availability

The original contributions presented in this study are included in the manuscript. Further inquiries may be directed to the corresponding author.

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
