# Peer review of "Custom-Designed Portable Potentiostat and Indirect Cyclic Voltammetry Index Analysis for Rapid Pesticide Detection Using Molecularly Imprinted Polymer Sensors"

_sensors, 2025, doi:10.3390/s25102999_

Round 1
Reviewer 1 Report (New Reviewer)
Comments and Suggestions for Authors
The authors present a miniature portable electrochemical analysis platform based on cyclic voltammetry for rapid detection of pesticides. The system was tested for its relevance in comparison with commercial electrochemical instruments. However, there are serious errors in the manuscript and acceptance is not recommended. The comments are as follows.
- Comparing the initial current values of NCIR and CHI in the detection of pesticides at low concentrations (0.01 M and 0.02 M) in Fig. 6 does not agree with the contents of the manuscript.
- Show in Fig. 6 the reason for the appearance of a pair of redox peaks in the detection of chlorpyrifos and elaborate on the mechanism.
- Explain the reason for the appearance of a small oxidation peak at 0.5 V in Fig. 7.
- Why the authors chose to test at a step size of 5 mV resulting in insufficient electron transfer kinetics and whether larger step sizes were tried.
- The concentration of 0.01 M electrolyte solution was chosen to smooth out the electrical signal, but this increases the solution resistance. Could this affect the current response at low concentrations of pesticides.
- The correlation coefficient (R2) in Figure 8 is too low, and consideration needs to be given to whether there is a correlation between concentration and total area.
7. There is a serious error in Figure 9, please check.
Author Response
Reviewer#1
The authors present a miniature portable electrochemical analysis platform based on cyclic voltammetry for rapid detection of pesticides. The system was tested for its relevance in comparison with commercial electrochemical instruments. However, there are serious errors in the manuscript and acceptance is not recommended. The comments are as follows.
1. Comparing the initial current values of NCIR and CHI in the detection of pesticides at low concentrations (0.01 M and 0.02 M) in Fig. 6 does not agree with the contents of the manuscript.
Response: Thanks for the reviewer’s suggestion. Figure 6 presents a comparison of the performance between the NCIR system and the CHI system, using potassium hexacyanoferrate(II) trihydrate (K₄[Fe(CN)₆]·3H₂O) combined with potassium ferricyanide (K₃Fe(CN)₆) as the electrolyte. This experiment evaluates the response of both systems across different electrolyte concentrations, focusing on their electrochemical behavior. It should be noted that this comparison pertains to the initial current values of the electrolyte solutions, rather than pesticide detection.
2. Show in Fig. 6 the reason for the appearance of a pair of redox peaks in the detection of chlorpyrifos and elaborate on the mechanism.
Response: Thanks for the reviewer’s suggestion. As the test samples in Figure 6 consist of potassium hexacyanoferrate(II) trihydrate (K₄[Fe(CN)₆]·3H₂O) combined with potassium ferricyanide (K₃Fe(CN)₆) electrolyte, characteristic redox peaks are observed in the voltammograms. It should be clarified that these results are not derived from chlorpyrifos samples.
3. Explain the reason for the appearance of a small oxidation peak at 0.5 V in Fig. 7.
Response: Thanks for the reviewer’s suggestion. The appearance of a minor anodic peak around 0.5 V may be attributed to the complex nature of the pesticide, whose molecular structure potentially interacts with the electrode surface, resulting in reversible or quasi-reversible redox processes. Additionally, variations in electrolyte composition, fluctuations in pH, and the presence of trace impurities could alter the electrochemical environment, thereby contributing to the emergence of secondary oxidation peaks within the cyclic voltammogram.
4. Why the authors chose to test at a step size of 5 mV resulting in insufficient electron transfer kinetics and whether larger step sizes were tried.
Response: Thanks for the reviewer’s suggestion. In the circuit design and experimental process, the primary consideration was to compare the time required for each cycle within the voltage range of -1 V to 1 V, aiming to achieve a measurement duration comparable to that of the CHI system. When the step size is excessively small, the time consumed per cycle increases significantly. Conversely, employing a larger step size substantially shortens the measurement time but compromises the resolution of the acquired data. Therefore, a step size of 5 mV was selected as an optimal compromise, balancing measurement duration and resolution for reliable and efficient analysis.
5. The concentration of 0.01 M electrolyte solution was chosen to smooth out the electrical signal, but this increases the solution resistance. Could this affect the current response at low concentrations of pesticides.
Response: Thanks for the reviewer’s suggestion. The experiment of using 0.01M electrolyte with pesticide samples is only used to find better characteristic indices. In the low-concentration experiments, ultrapure water was used as the background solution, which had little effect on low-concentration pesticides.
6. The correlation coefficient (R2) in Figure 8 is too low, and consideration needs to be given to whether there is a correlation between concentration and total area.
Response: Thanks for the reviewer’s suggestion. Based on the experimental results, under the current conditions, the relatively low correlation between sample concentration and total voltammetric area may be attributed to the response and sensitivity of the MIP towards the pesticide analyte. Future work will focus on optimizing the design of the MIP and refining the electrode modification process, with the aim of enhancing the relationship between sample concentration and total CV area, thereby improving the sensor’s quantitative performance.
7. There is a serious error in Figure 9, please check.
Response: Thanks for the reviewer’s suggestion. We modified the X-axis and Y-axis descriptions of Figure 9.

Reviewer 2 Report (New Reviewer)
Comments and Suggestions for Authors
After reviewing the manuscript entitled “Miniaturized Electrochemical Sensor Platform for Precise and Rapid Pesticide Detection in Environmental Monitoring”, I address the following suggestions to the authors to increase the quality of the work:
- The authors have written an introduction similar to a literature review, with little focus on the main themes of the work. I recommend a review with more commentary on the analyte used in the proof of concept, on the development of portable potentiostats, on the MIP used in this work and, in addition, more emphasis on the central objectives of the manuscript.
- The authors have added an exclusive section on “Working principles”, however when presenting the data there are no numerical values. I recommend that the currents, areas and potentials be discussed and presented more clearly.
- The commercial potentiostat used to compare the results must have its make and model presented in the “Methods and Materials” section. Data on the screen-printe electrode used should also be presented. How was it constructed? What is the composition of the inks? The authors should also provide more information about the MIP such as: synthesis reaction and reagents used, its characterization, structure, among others.
- I recommend that the authors standardize the concentrations and units presented. In the manuscript, units such as mol/L, % and ppm were used.
- With regard to the comparison of the sensor's performance, some points should be better elucidated. In Figure 6 the cyclic voltammograms indicate a reversibility of the ferri/ferrocyanide probe ranging from -0.5 to +0.5 V, while in Figure 7 this value improves considerably. Why? Were different electrodes used?
- Furthermore, why did the authors opt for an indirect interpretation of the oxidation-reduction process of chlorpyrifos? There are references in the literature (10.1016/j.jiec.2022.05.007, 10.1016/j.electacta.2023.143305, 10.1039/D4RA04406A) that indicate processes taking place at potentials close to 0.1 V vs Ag|AgCl|KCl(sat).
- In Figure 10 there is a considerable loss of resolution in the voltammograms. What is the reason for this? Could this be related to the choice of using ultrapure water as the supporting electrolyte? Why wasn't a buffer solution used?
- Determining the limit of detection is confusing. I recommend the authors present a calibration curve and derive the LOD from the curve parameters.
- In lines 238-241 the authors make statements about the system developed having advantages over commercial ones. I disagree. There are currently portable potentiostats for field analysis with reduced sizes, wireless connection and integration with smartphones. The area for software and hardware development is interesting, but we cannot believe that what is available commercially is limited and bad.
- I recommend that the title be reconsidered. The work focuses on the development of a low-cost potentiostat, I think these terms might make the work more attractive. Also, more information about the process of building and programming the platform should be included. What is the effective cost of this system?
Author Response
After reviewing the manuscript entitled “Miniaturized Electrochemical Sensor Platform for Precise and Rapid Pesticide Detection in Environmental Monitoring”, I address the following suggestions to the authors to increase the quality of the work:
- The authors have written an introduction similar to a literature review, with little focus on the main themes of the work. I recommend a review with more commentary on the analyte used in the proof of concept, on the development of portable potentiostats, on the MIP used in this work and, in addition, more emphasis on the central objectives of the manuscript.
Response: Thanks for the reviewer’s suggestion. The analytes employed in this study, the development of the portable potentiostat, and the MIP utilized are primarily described in Section 2.3. Additionally, the central objective of this work are specifically addressed in the final paragraph of the Introduction.
- The authors have added an exclusive section on “Working principles”, however when presenting the data there are no numerical values. I recommend that the currents, areas and potentials be discussed and presented more clearly.
Response: Thanks for the reviewer’s suggestion. We have revised the original "Working Principle" section by integrating its content into a newly structured Materials and Methods chapter. This section is now systematically organized according to the employed theoretical basis (cyclic voltammetry), the molecularly imprinted polymer (MIP) technique, experimental setup and sample preparation, as well as detailed descriptions of the experimental procedures.
- The commercial potentiostat used to compare the results must have its make and model presented in the “Methods and Materials” section. Data on the screen-printe electrode used should also be presented. How was it constructed? What is the composition of the inks? The authors should also provide more information about the MIP such as: synthesis reaction and reagents used, its characterization, structure, among others.
Response: Thanks for the reviewer’s suggestion. We added the details of the screen-printed electrode, composition of inks, information about the MIPs. The screen-printed carbon electrode (SPCE) comprises a working electrode and a counter electrode fabricated using carbon-based conductive ink, along with a reference electrode printed with silver/silver chloride (Ag/AgCl) ink. The electrode layout is defined and insulated by a polymeric layer to ensure precise control over the active surface areas. The working electrode features a circular geometry with a diameter of 3.35 mm. Functional monomers (methacrylic acid), cross-linker (ethylene glycol dimethacrylate), and initiator (azobisisobu-tyronitrile) were prepared to create the MIPs with the chlorpyrifos acting as the target molecule. Please find the description in Section 2.3 in the revised manuscript.
- I recommend that the authors standardize the concentrations and units presented. In the manuscript, units such as mol/L, % and ppm were used.
Response: Thanks for the reviewer’s suggestion. We modified the mol/L and % into ppm in the experiment and results sections of the revised manuscript.
- With regard to the comparison of the sensor's performance, some points should be better elucidated. In Figure 6 the cyclic voltammograms indicate a reversibility of the ferri/ferrocyanide probe ranging from -0.5 to +0.5 V, while in Figure 7 this value improves considerably. Why? Were different electrodes used?
Response: Thanks for the reviewer’s suggestion. The reason is, in Figure 6, the samples consist solely of electrolytes at varying concentrations, whereas in Figure 7, the test samples involve a fixed concentration of electrolyte combined with different concentrations of chlorpyrifos pesticide solution. The presence of the pesticide may influence the electrochemical reactions, thereby affecting the observed response. Both experiments were conducted using the same electrode configuration.
- Furthermore, why did the authors opt for an indirect interpretation of the oxidation-reduction process of chlorpyrifos? There are references in the literature (10.1016/j.jiec.2022.05.007,10.1016/j.electacta.2023.143305, 10.1039/D4RA04406A) that indicate processes taking place at potentials close to 0.1 V vs Ag|AgCl|KCl(sat).
Response: Thanks for the reviewer’s suggestion. In this study, an indirect interpretation of the redox behavior of chlorpyrifos was initially adopted due to the considerable complexity of its electrochemical response when interacting with the modified electrode and the matrix environment. The electrochemical signals were influenced not only by the intrinsic properties of the pesticide but also by the MIP layer and potential secondary processes occurring within the electrolyte system.
Recent studies have indeed demonstrated the direct electrochemical processes of chlorpyrifos. These references are highly relevant; however, under our specific experimental configuration and with the constraints of our portable system, it was challenging to establish a direct correlation between the CV profiles and pesticide concentration. Therefore, we prioritized the use of indirect indices as a more practical and reliable approach in this context.
Future work will focus on refining the electrode modification strategies and optimizing the detection environment, with the aim of further investigating and clarifying the direct redox signatures of chlorpyrifos, particularly within the low-potential region, to enhance both sensitivity and interpretability of the results.
- In Figure 10 there is a considerable loss of resolution in the voltammograms. What is the reason for this? Could this be related to the choice of using ultrapure water as the supporting electrolyte? Why wasn't a buffer solution used?
Response: Thanks for the reviewer’s suggestion. Due to the difficulty of using an electrolyte as a background solution in outdoor experiments, the water was subsequently employed as a substitute for background solution, and the ultrapure water was used as the background solution in the experiment. If saturation of the curve occurs during measurement, the resistance in NCIR module is switched to a smaller value, and it will expand the measurement range but reduce the measuring resolution, resulting in a step-like appearance of the CV curve However, this setting enables a faster measurement process. In future studies, we will refine our measurement circuitry to enhance resolution and improve data accuracy.
- Determining the limit of detection is confusing. I recommend the authors present a calibration curve and derive the LOD from the curve parameters.
Response: Thanks for the reviewer’s suggestion. We acknowledge that the current description of the limit of detection (LOD) determination may lack clarity. In future research, we will include a well-defined calibration curve based on the relationship between analyte concentration and the corresponding electrochemical response. The LOD will be calculated using the standard approach derived from the calibration curve parameters, typically as LOD = 3σ / S, where σ represents the standard deviation of the blank measurements and S is the slope of the calibration curve. This will provide a more rigorous and transparent assessment of the system's detection capability.
- In lines 238-241 the authors make statements about the system developed having advantages over commercial ones. I disagree. There are currently portable potentiostats for field analysis with reduced sizes, wireless connection and integration with smartphones. The area for software and hardware development is interesting, but we cannot believe that what is available commercially is limited and bad.
Response: Thanks for the reviewer’s suggestion. We know your point regarding the current availability of highly advanced commercial portable potentiostats, many of which indeed feature compact designs, wireless connectivity, and smartphone integration for field applications. Our intention was not to undervalue the capabilities of existing commercial systems, but rather to highlight the specific adaptability and customization advantages of our system within the context of this study-particularly in terms of open-source software integration, tailored measurement protocols, and cost-effectiveness for targeted pesticide sensing applications. We agree that the field of portable potentiostat development has progressed significantly, and our system as a complementary, application-specific alternative rather than a superior general-purpose solution. We modified the word "Unlike commercial instrument" into "Unlike high-end, precision commercial analytical instruments", which can emphasize our advantages of customization and low cost.
- I recommend that the title be reconsidered. The work focuses on the development of a low-cost potentiostat, I think these terms might make the work more attractive. Also, more information about the process of building and programming the platform should be included. What is the effective cost of this system?
Response: Thanks for the reviewer’s suggestion. We have modified the title into “Custom-Designed Portable Potentiostat and Indirect CV Index Analysis for Rapid Pesticide Detection Using MIP Sensors”. The cost of the system was about USD 500-1,000. The details of the device and platform were described in the Section 2.3 in the revised manuscript.

Round 2
Reviewer 1 Report (New Reviewer)
Comments and Suggestions for Authors
The results and discussion is not sufficient and accurate. Please confirm the R2 value in Fig. 9.
If no problem, I recommend acceptance of the paper.
Author Response
Reviewer#1
- The results and discussion is not sufficient and accurate. Please confirm the R2 value in Fig. 9.
Response: Thanks for the reviewer’s suggestion. After reconfirming the experimental data, the potential at the cathodic peak (Epc ) shows no appreciable linear dependence on concentration. The values fluctuate only within the experimental noise band, the linear regression explains virtually none of the variance—hence R² ≈ 0.003. In contrast, the cubic polynomial has four adjustable coefficients that can be forced to pass through every point, thus driving R² close to 1.0. Therefore, the low R² of the linear fit indicates that Epc is essentially independent of concentration under the current conditions. Future work will focus on optimizing the design of the MIP and refining the electrode modification process, with the aim of enhancing the relationship between sample concentration and total CV area and potential corresponding to peak reduction current, thereby improving the sensor’s quantitative performance.

Reviewer 2 Report (New Reviewer)
Comments and Suggestions for Authors
After a first round of revision, the manuscript improved considerably in quality, with a more detailed description of the methods and data obtained. Two questions can still be improved and answered:
1. I recommend that the authors subdivide Section 2.3 into each of the topics mentioned in it (Sensor and potentiostat, Electrochemical measurements, MIP synthesis and modification).
2. The concentration of pesticide used was 7516 ppm, which corresponds to approximately 21.44 mmol/L? These values are quite high, check that they are correct.
Author Response
Reviewer#2
After a first round of revision, the manuscript improved considerably in quality, with a more detailed description of the methods and data obtained. Two questions can still be improved and answered:
- I recommend that the authors subdivide Section 2.3 into each of the topics mentioned in it (Sensor and potentiostat, Electrochemical measurements, MIP synthesis and modification).
Response: Thanks for the reviewer’s suggestion. We have added the sub-topics included the Potentiosat and Sensor, Electrochemical measurements, MIP synthesis and modification, Pesticide in the revised manuscript.
- The concentration of pesticide used was 7516 ppm, which corresponds to approximately 21.44 mmol/L? These values are quite high, check that they are correct.
Response: Thanks for the reviewer’s suggestion. After a meticulous reassessment of the molecular formulae and molar masses, excluding the hydrate’s water content and re-calculating gravimetrically, the electrolyte’s concentration was revised from 7516 ppm to approximately 3476 ppm, corresponding to 0.01 mol/L (M).

This manuscript is a resubmission of an earlier submission. The following is a list of the peer review reports and author responses from that submission.
Round 1
Reviewer 1 Report
Comments and Suggestions for Authors
1. In the cyclic voltammetry analysis, the CV curves obtained for different concentrations of the pesticide exhibit a pulse-like signal rather than the expected curve behavior. Could the authors provide an explanation for this observation?
2. Please recheck all the notations used in the manuscript.
3. How did the author calculate the limit of detection (LOD)? Please mention the equation used.
4. Please ensure that the notation for Pb2+ is properly formatted throughout the manuscript.
5. Recommend to providing a more detailed discussion of Figure 6, particularly the peak-to-peak separation.
Reviewer 2 Report
Comments and Suggestions for Authors
Comments to authors
- “Traditional methods for detecting concentrations of pesticide, organic compounds, or heavy metals primarily rely on professional laboratory equipment such as liquid chromatography, composition analyzers, and electrochemical analyzer. These methods offer advantages of high accuracy, but the disadvantages include the inability to provide results in real time, the need for highly skilled operators and expensive equipment. This study introduces a miniaturized and portable electrochemical analyzer and characteristic indices platform, capable of rapidly detecting pesticide concentration based on cyclic voltammetry (CV)”. This statement is contradictory, the authors discussed the disadvantages of the electrochemical analyzer but still chose to use it.
- “The total area of the CV curve exhibited a linear correlation with pesticide concentration, achieving an R² value of 0.89”. This is too low to use the corresponding linear plot to estimate the concentration of the analyte. At least 0.99 should be targeted in order to accurately quantify the analyte.
- The Abstract should be re-written, it does not give much information regarding what was done and the results obtained.
- Which pesticide is being studied? What is its structure? What is the chemistry behind its detection using CV? These are critical questions that the manuscript does not respond to.
- The introduction should be improved and focus on the analyte of interest. It should be reduced as well to a maximum of 2 pages. This will enable authors to focus more on their own results.
- The manuscript is written like a review paper, it does not give detailed information on the experimental aspects of the study. Thus, the manuscript lacks experimental results to support the findings presented.
- The coefficient of determination in Figure 8 is too low for quantification purposes. These should be revised accordingly as it is recommended that the coefficient of determination should be close to 1.
- The CV in figure 10 should be repeated as it seems like there were interferences during measurements.
- The complete analytical figures of merits such as the linear range, limit of quantification, interday and intraday precision, sensitivity, amongst others should be evaluated.
- Interference studies should be carried out to ensure that the analyte is not interfering with any other analyte.
- Which environmental samples is the pesticide of interest analyzed from? What was the initial concentration of the pesticide in the samples? Was the spiking done and what concentrations were detected?
- Other parameters besides the concentration such as pH and Mass of the MIPs, amongst others, should be studied as they affected the analyte’s detection.
- Other electrochemical techniques such as square wave voltammetry (SWV), differential pulse voltammetry (DPV) and chronoamperometry, should be compared and contrasted with the current CV technique in a Table format concerning the pesticide of interest.
- Which Molecularly imprinted polymers are used as recognition elements? This is not clearly discussed in the manuscript.
Reviewer 3 Report
Comments and Suggestions for Authors
Some suggestions to improve the investigation and manuscript:
line 24 – first-time introduced abbreviation should be spelled out (MIP).
lines 37-39 – The statistical data about the amount of people undergo the contaminated water should be accompanied with references.
line 156 – sensor
The Introduction section should be rewritten and shortened. It should be divided on paragraphs that would introduce the readers to the chosen scientific field. It should not be a “review-like” paper. There is no need to detailed description of all papers being referred. The aforementioned paragraphs should be aimed at the scientific problem formulation; the current approaches to solve this problem, including advantages and disadvantages (instead of Table 1); electrochemical sensors with classification on, for instance, the modifying layer composition (not only MIP sensors) and/or transducer’s material; brief description of suggested approach. The number of references should be increased, especially where the application of so-called “indices” is discussed.
lines 180, 208 – The term “potential voltage” is incorrect as well as the statement about the direct measure of analyte's concentration in solution (lines 216-217). There are two basic equations for the diffusion-free redox probes: Cottrell equation for chronoamperometry and Randles–Sevcik equation for cyclic voltammetry.
line 208 – corresponding
The Figure 1 is incorrect in the way of peak currents calculation and anodic/cathodic peak positions. Authors should recheck the ref. [19].
lines 259, 283 – What is KCN3/4? Is it potassium cyanide or Fe(CN)63-/4-? Both of them are not a background electrolyte. The commonly used background electrolytes in electrochemical investigations are potassium/sodium nitrate/sulfate/chloride. 0.1 M concentration of electrolyte is considered as appropriate. Low conductivity (high ohmic resistance) of working solution can be recognized by the CV shape (Fig. 6).
Cypermethrin is denoted as template (line 275) and chlorpyrifos as well (line 278). There are no any results for Cypermethrin in the manuscript.
The using of MIP in electrochemical sensors involves the comparison with results obtained for the NIP electrodes.